# The impact of the National Programme to Eliminate Lymphatic Filariasis on filariasis morbidity in Sri Lanka: Comparison of current status with retrospective data following the elimination of lymphatic filariasis as a public health problem

**Indeewarie E Gunaratna[1], Nilmini T. G. A Chandrasena[2]\*, Murali Vallipuranathan[1], Ranjan Premaratna[3], Dileepa Ediriweera[4], Nilanthi R de Silva[2]**

**1** Anti Filariasis Campaign, Ministry of Health, Colombo, Sri Lanka, **2** Department of Parasitology, Faculty of Medicine, University of Kelaniya, Kelaniya, Sri Lanka, **3** Department of Medicine, Faculty of Medicine, University of Kelaniya, Kelaniya, Sri Lanka, **4** Health Data Science Unit, Faculty of Medicine, University of Kelaniya, Kelaniya, Sri Lanka

\* nilmini@kln.ac.lk

## Abstract

### Introduction

Sri Lanka implemented the National Programme for Elimination of Lymphatic Filariasis (NPELF) in its endemic regions in 2002. Five annual rounds of mass drug administration using the two-drug combination diethylcarbamazine (DEC) and albendazole led to sustained reductions in infection rates below threshold levels. In 2016, WHO validated that Sri Lanka eliminated lymphatic filariasis as a public health problem.

### Objective

To explore the impact of the NPELF on lymphatic filariasis morbidity in Sri Lanka.

### Methods

Passive Case Detection (PCD) data maintained in filaria clinic registries from 2006–2022 for lymphoedema and hospital admission data for managing hydroceles/spermatoceles from 2007–2022 were analyzed. The morbidity status in 2022 and trends in overall and district-wise PCD rates were assessed. Poisson log-linear models were used to assess the trends in PCD for endemic regions, including district-wise trends and hospital admissions for the management of hydroceles/spermatoceles.

### Results

In 2022, there were 566 new lymphoedema case visits. The mean (SD) age was 53.9 (16.0) years. The staging was done for 94% of cases, of which 79% were in the early stages

**Data Availability Statement:** All relevant data are included in the manuscript. In addition the current data (2022 and 2023) may be accessed via the Annual Health Bulletin of AFC ;https://afc.health. gov.lk/annual-reports-health-bulletin/ at the AFC website, https://afc.health.gov.lk/.

**Funding:** The author(s) received no specific funding for this work.

**Competing interests:** The authors have declared that no competing interests exist.

(57.3% and 21.4% in stages two and one, respectively). Western Province had the highest caseload (52%), followed by the Southern (32%) and Northwestern (16%) Provinces, respectively. The reported lymphoedema PCD rate in 2022 was 0.61 per 10,000 endemic population. The overall PCD rate showed a decline of 7.6% (95%CI: 4.9% - 10.3%) per year (P < 0.0001) from 2007 to 2022. A steady decline was observed in Colombo, Gampaha and Kurunegala districts, while Kalutara remained static and other districts showed a decline in recent years. Further, admissions for inpatient management of hydroceles/spermatoceles showed a declining trend after 2015.

## Conclusions

The PCD rates of lymphoedema and hydroceles/spermatoceles showed a declining trend in Sri Lanka after the implementation of the NPELF.

### Author summary

In the year 2000, the World Health Organization initiated the mass drug administration program to eliminate lymphatic filariasis. Sri Lanka implemented the program for five years (2002–2006), offering yearly single doses of anti-filarial treatment to the eligible population of all eight endemic districts. In 2016, Sri Lanka was acknowledged as having eliminated lymphatic filariasis as a public health problem. This study explored the impact of the mass drug administration program on lymphoedema and hydrocele/spermatoceles passive case detection rates using filaria clinic and hospital admission data. A reduction in new lymphoedema case visits to filaria clinics and hospital admissions for hydrocele/spermatoceles management was noted two decades after implementing mass anti-filarial treatment in Sri Lanka.

## Introduction

Lymphatic filariasis (LF), ranked as one of the world's leading causes of permanent and long-term disability, is targeted for global elimination [1,2]. The causative nematodes, *Wuchereria bancrofti*, *Brugia malayi* and *B. timori* dwell within the lymphatic system (distal lymphatics and lymph nodes) and impair lymph drainage, causing severely disabling chronic manifestations, lymphoedema and its severe form elephantiasis of the legs and hands and hydrocoele [3]. In bancroftian filariasis, lymphoedema and elephantiasis commonly affect the lower legs and thighs, while less commonly, other sites such as arms, scrotum, penis, vulva, and breasts are affected, whereas in *B. malayi* infections, the lymphoedema is confined to below the knees and genito urinary manifestations such as hydrocoele and chyluria are a rarity [4]. Considering the huge socioeconomic burden of LF along with advances in diagnostics and treatment, the World Health Assembly in 1997 resolved to eliminate LF as a public health problem [5]. Subsequently, the World Health Organization launched the Global Programme to Eliminate LF in the year 2000 based on twin strategies: annual mass drug administration (MDA) to all at-risk populations and managing morbidity and preventing disability (MMDP) [6]. In the roadmap for eliminating neglected tropical diseases (NTDs), the global elimination of LF as a public health problem is targeted for 2030 [7].

In line with the global initiative, Sri Lanka's National Programme for Elimination of Lymphatic Filariasis (NPELF) was implemented in 2002 in the eight endemic districts: Colombo, Gampaha and Kalutara of Western Province, Galle, Matara and Hambantota of Southern

Province and Kurunegala and Puttalam of Northwestern Province. Bancroftian filariasis was highly endemic in the densely populated towns of Colombo, Galle and Matara, where unplanned urbanization and coconut coir industry in Galle and Matara provided abundant breeding sites for the principal vector *Culex quinquefasciatus* while brugian filariasis was considered eliminated at that time [8]. Five annual rounds of MDAs using DEC and albendazole were administered in the endemic region from 2002–2006. MMDP services were provided to the diseased via a network of filaria clinics. In 2016, Sri Lanka was declared to have eliminated LF as a public health problem after meeting all the criteria stipulated by the WHO to verify elimination [9].

Since initiating the national elimination program, the LF infection parameters have shown a steady decline. The overall microfilaria (MF) rate, which was 0.2% in 2001, before the launch of the NPELF, was down to 0.06% in 2016, at the time of declaration of elimination as a public health problem and is 0.03% at present [8]. Thus, two decades following the elimination drive, the percentage reduction of microfilaraemia was 80%. The vector infection and infective rates have declined to 0.49% and 0.03%, respectively [10]. A limited number of studies have been conducted to date in Sri Lanka to examine the impact of the NPELF on chronic LF morbidity states (lymphoedema and hydrocoele) [11,12]. Worldwide, few have examined the impact of MDA from a clinical perspective, and reports are inconsistent [13].

At present, Sri Lanka is in the LF post-validation surveillance phase with its main objective of reducing the reservoir of residual infections in the population, which involves detecting, mapping and treating MF carriers to reduce the risk of transmission and prevent the re-establishment of infection. Thus, priority is given to parasitological and entomological surveillance. The nationwide burden of LF morbidity has never been assessed, and no active disease surveillance program exists. However, the network of filaria clinics scattered within the endemic regions maintains registries of patients who present directly or are referred for lymphoedema management. These filaria clinics provide the basic morbidity care package which includes advice and demonstration on the significance of daily washing of the affected areas in the prevention of Acute Dermato-Lymphangio-Adenitis (ADLA), patient education on skincare and hygiene, provision of recommended antibacterials and antiseptics to minimize the ADLA episodes triggering the progression of lymphoedema and promotion of measures to improve lymph drainage (compression, elevation and exercises) [14]. The MMDP services in Sri Lanka do not encompass tertiary care (rehabilitation, psychological and social support).

Patients with hydroceles rarely visit the filaria clinics as these clinics are not equipped to provide surgical services. Thus, data on hydrocele cases are rarely documented in the filaria clinic registries; however, government hospital admissions for hydrocele management are documented in state hospitals' Indoor Morbidity and Mortality Registers.

This study aimed to examine the impact of the NPELF on LF chronic disease manifestations using Passive Case Detection (PCD) data maintained in filaria clinics and hospital indoor morbidity and mortality registers. The trends in lymphoedema PCD rates were examined (overall and district-wise) by a retrospective analysis of clinic data since initiating the LF elimination program in Sri Lanka to identify and anticipate disease patterns and directions. The hospital inpatient records for hydrocele/spermatocele management from 2007–2022 throughout the country were examined for trends.

## Methods

The Ethics Review Committee of the Faculty of Medicine, University of Kelaniya granted ethics exemption for the study. The study was an audit of data archives and current data (available in the public domain) of the National Anti-Filariasis Campaign.

### Analysis of current status

The lymphoedema case attendance at filaria clinics in 2022 for the eight endemic districts in three provinces was obtained from the recently established online platform at Anti-Filariasis Campaign and analyzed to assess the current disease status. For 2022, the overall and district-wise lymphoedema PCD rate per 10,000 population was calculated using the estimated respective populations [15].

### Retrospective analysis of PCD trends

A retrospective analysis was conducted using the data maintained in the filaria clinic registries from 2006 onwards. We extracted data on clinic attendees focusing on the first clinic visits (new lymphoedema case visits) to filaria clinics from 2006 to 2022, which served as a proxy for disease incidence. Information regarding the estimated population in the endemic districts for the relevant years was obtained from the national censuses of 2001 and 2012 [15,16]. The age distribution of new lymphoedema cases was analyzed using the available data from 2019 onwards. The district-wise clinic attendance data was available from 2012 onwards, and choropleth maps were used to visualize the district-wise reported lymphoedema PCD rates.

Separate Poisson log-linear models were used to assess the overall and district-wise trends in PCD rates. The number of PCD cases was used as the response variable, and the total respective population of interest was considered a log offset in the models. Time since 2006 was considered as the explanatory variable in the overall model, and time since 2012 was considered the explanatory variable in the district-wise models. Piece-wise and quadratic functions for a time were considered appropriate to model non-linear trends. All the fitted models showed overdispersion, and quasi-Poisson models were adopted.

The annual country-wide data on state hospital admissions for the management of hydrocele/spermatocele from 2007 to 2022 was extracted from the indoor morbidity and mortality registers. The Poisson log-linear model was used to assess the trends in hospital admissions, considering the number of admissions as the response variable and the total male population as a log offset in the models. The time since 2007 was considered as the explanatory variable with a piece-wise function to model non-linear trends. The fitted model showed overdispersion, and the quasi-Poisson model was adopted.

A P value $< 0.05$ was considered as significant. R programming language 4.4.0 was used for analysis.

## Results

### New lymphoedema case presentations in 2022 in the endemic region

In 2022, 566 patients with lymphoedema received MMDP services for the first time at Filaria clinics (i.e. first clinic visits). The mean age of these patients was 53.9 (SD 16.0) years. Three percent (n = 15) of those seeking care were less than 21 years of age. The demography and disease severity of the new clinic attendees is given in Table 1. Data on lymphoedema severity was documented for 94%, of which 79% were in the early stages (57.3% and 21.4% in stages two and one, respectively). Most (52%,) were from the Western Province (Colombo 108, Gampaha 52, Kalutara 113), while 32% were from Southern (Galle 30, Matara 74, Hambantota 27) and 16% from Northwestern (Kurunegala 98, Puttalam 63) Provinces. In 2022, the reported overall PCD rate for lymphoedema was 0.61 per 10,000 in the endemic population, while the districts of Matara (1.11 per 10,000), Puttalam (0.78 per 10,000) and Kalutara (0.68 per 10,000) reported the highest PCD rates.

**Table 1. Demography and disease severity of lymphoedema case attendees visiting the filaria clinics for the first time (new clinic visits) in 2022.**

| Variables | Western Province | | | Southern Province | | | North Western Province | | Non-endemic | Total |
|---|---|---|---|---|---|---|---|---|---|---|
| | Colombo | Gampaha | Kalutara | Galle | Matara | Hambantota | Kurunegala | Puttalam | | N (%) |
| **Age group** | | | | | | | | | | |
| <21 years | 01 | 04 | 01 | 00 | 00 | 00 | 07 | 03 | 00 | 16 (2.8) |
| 21–30 years | 06 | 00 | 08 | 00 | 02 | 01 | 07 | 04 | 01 | 29 (5.1) |
| 31–40 years | 08 | 06 | 11 | 02 | 06 | 05 | 12 | 11 | 00 | 61 (10.8) |
| 41–50 years | 27 | 14 | 26 | 08 | 07 | 09 | 19 | 12 | 00 | 122 (21.6) |
| 51–60 years | 23 | 09 | 26 | 04 | 13 | 05 | 18 | 13 | 00 | 111 (19.6) |
| 61–70 years | 29 | 12 | 29 | 10 | 26 | 05 | 18 | 16 | 00 | 145 (25.6) |
| >70 years | 12 | 07 | 12 | 06 | 16 | 00 | 16 | 04 | 00 | 74 (13.1) |
| Unspecified | 02 | - | - | - | 04 | 02 | - | - | - | 08 (1.4) |
| **Sex** | | | | | | | | | | |
| Female | 59 | 23 | 55 | 12 | 40 | 12 | 49 | 36 | 00 | 286 (50.5) |
| Male | 48 | 29 | 58 | 18 | 32 | 14 | 49 | 27 | 01 | 276 (48.8) |
| Unspecified | 01 | - | - | - | 02 | 01 | - | - | - | 04 (0.7) |
| **Stage** | | | | | | | | | | |
| I | 15 | 11 | 21 | 04 | 29 | 07 | 08 | 18 | 01 | 114 (20.1) |
| II | 70 | 32 | 64 | 09 | 15 | 14 | 63 | 38 | 00 | 305 (53.9) |
| III | 15 | 07 | 16 | 05 | 07 | 04 | 24 | 03 | 00 | 81 (14.3) |
| IV | 01 | 01 | 03 | 05 | 01 | 00 | 02 | 00 | 00 | 13 (2.3) |
| V | 00 | 00 | 04 | 02 | 01 | 00 | 00 | 01 | 00 | 08 (1.4) |
| VI | 01 | 01 | 00 | 02 | 00 | 00 | 01 | 02 | 00 | 07 (1.2) |
| VII | 00 | 00 | 00 | 03 | 01 | 00 | 00 | 00 | 00 | 04 (0.7) |
| Unspecified | 06 | - | 05 | - | 20 | 02 | - | 01 | - | 34 (6.0) |
| **Total** | **108** | **52** | **113** | **30** | **74** | **27** | **98** | **63** | **01** | **566 (100)** |

## Retrospective analysis of new lymphoedema cases

The reported overall PCD rate in the endemic districts in 2006 was 2 per 10,000, which had declined to 0.6 per 10,000 in 2022. Fig 1 shows the endemic districts for filariasis and changes in district-wise reported PCD rates of lymphoedema from 2012 to 2022. The observed PCD rates showed different time trends in the seven endemic districts.

The fitted log-linear models for overall PCD rates from 2006 to 2022 and district-wise PCD rates from 2012 to 2022 are shown in Table 2. The overall PCD rate had declined from 2006 to 2022 at a rate of 7.6% (95%CI: 4.9% - 10.3%) per year (P < 0.0001). The estimated overall PCD rate from the fitted model for 2022 was 0.45 (95%CI: 0.33–0.61) per 10,000. The district-wise PCD rates showed declining trends in three of the seven endemic districts: 14.5% (10.3% - 18.7%) per year in Colombo, 6.9% (1.4% - 12.3%) per year in Gampaha and 12.0% (8.3% - 17.3%) per year in Kurunegala. Puttalam and Galle showed quadratic trends; PCD rates rose till 2018 in Puttalam and till 2017 in Galle, and rates went down thereafter. In Matara, PCD rose till 2014, and rates showed a slight downward trend thereafter. The PCD rate remained static in the Kalutara district (Fig 2). The Hambantota district was excluded from the analysis as it did not have MMDP coverage until 2022. The age distribution of new lymphoedema cases throughout the endemic districts from 2019–2022 showed a rising frequency with age, with the average case presenting in the 6th decade (Table 3).

## Trends in the inpatient management of hydroceles/spermatoceles

A total of 37,557 patients were admitted for management for hydroceles/spermatoceles in state hospitals from 2007 to 2022. This was 2,717 in 2007 and 1,869 in 2022. The fitted log-linear model

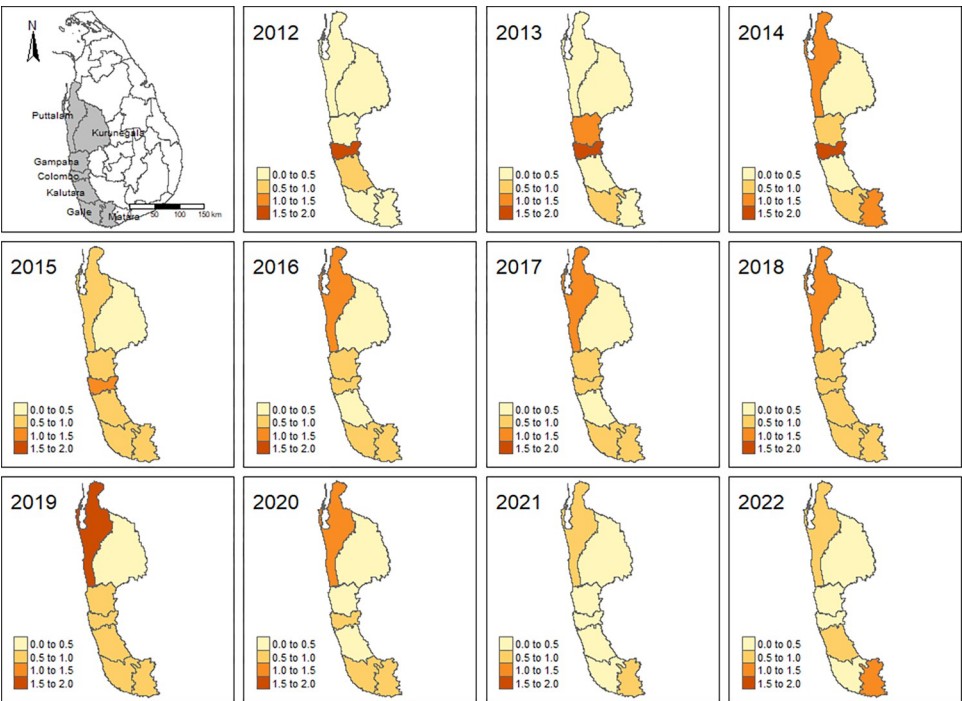

**Fig 1. The district-wise reported PCD rates of lymphoedema after implementing the national programme to eliminate lymphatic filariasis in Sri Lanka.** District-wise reported PCD rates of lymphedema from 2012 to 2022. The grey colour area in the top left map indicates the endemic districts of lymphatic filariasis in Sri Lanka. Subsequent maps show the district-wise PCD rates from 2012 to 2022. The base map shapefile was downloaded from The Humanitarian Data Exchange website (https://data.humdata.org/dataset/cod-ab-lka). The content on this site is licensed under a Creative Commons Attribution 4.0 International license (https://data.humdata.org/dataset).

for hospital admission for the management of hydroceles/spermatoceles is shown in Table 4. Hospital admissions remain static from 2007 to 2015 (P = 0.338). Thereafter, hospital admissions declined till 2022 at a rate of 7% per year (95% CI: 4% - 11% per year, P = 0.001) (Fig 3).

## Discussion

The focus of this study was to evaluate the impact of the mass treatment program on the burden of chronic lymphatic morbidity, from a clinical perspective rather than an epidemiological perspective of risk-of-infection (represented by MF rates, and mosquito infection and infective rates). During 2001–2019, the national program delivered a total of 52.8 million treatments of which 44.8 million treatments were consumed by a target population of 10.46 million [17].

Clinically, filarial lymphoedema is indistinguishable from lymphoedema due to other causes and the lack of a specific diagnostic tool to trace the etiology of lymphoedema and hydrocele, to filarial origin would have affected this assessment, the magnitude of which is unknown. The lack of baseline epidemiological data on morbidity and the absence of an active case surveillance program hindered achieving the study objectives to a greater extent. The impact of preventive chemotherapy on lymphatic morbidity was assessed utilizing passive case detection data.

The onset of lymphoedema in filariasis endemic areas generally occurs around puberty, and prevalence rises progressively with age [13]. The age distribution of first clinic-visit attendees reflects the epidemiological pattern of filarial lymphoedema, with case numbers progressively increasing with age (Tables 1 and 3). The disease distribution was similar among both sexes (Table 1). Most cases were in early lymphoedema (stages 2 and 1) according to WHO criteria

**Table 2. Parameter estimates of the fitted log-linear models.**

|  | Estimate | Std. Err. | T value | P value |
|---|---|---|---|---|
| Overall: from 2006 to 2022 |  |  |  |  |
| Intercept | -1.7837 | 0.117 | -15.688 | <0.0001 |
| Time (year) | -0.079 | 0.019 | -5.369 | <0.0001 |
| Colombo district: from 2012 to 2022 |  |  |  |  |
| Intercept | -1.565 | 0.1159 | -13.50 | <0.0001 |
| Time (year) | -0.157 | 0.0250 | -6.280 | 0.0001 |
| Gampaha district: from 2012 to 2022 |  |  |  |  |
| Intercept | -2.337 | 0.160 | -14.568 | <0.0001 |
| Time (year) | -0.073 | 0.030 | -2.412 | 0.0391 |
| Kalutara district: from 2012 to 2022 | -2.989 | 0.253 | -11.833 |  |
| Intercept | 5.143 | 85.920 | 0.060 | 0.9540 |
| Time (year) | -0.004 | 0.043 | -0.095 | 0.9260 |
| Kurunegala district: from 2012 to 2022 | -3.499 | 0.126 | -27.846 |  |
| Intercept | 273.793 | 52.759 | 5.190 | 0.0006 |
| Time (year) | -0.138 | 0.026 | -5.266 | 0.0005 |
| Puttalam district: from 2012 to 2022 |  |  |  |  |
| Intercept | -3.638 | 0.341 | -10.656 | <0.0001 |
| Time (year) | 0.587 | 0.133 | 4.411 | 0.0030 |
| Time$^2$ (year) | -0.049 | 0.012 | -4.194 | 0.0030 |
| Galle district: from 2012 to 2022 |  |  |  |  |
| Intercept | -1.151e+05 | 4.828e+04 | -2.384 | 0.044 |
| Time (year) | 1.141e+02 | 4.787e+01 | 2.384 | 0.044 |
| Time$^2$ (year) | -2.829e-02 | 1.187e-02 | -2.384 | 0.044 |
| Matara district: from 2012 to 2022 |  |  |  |  |
| Intercept | -5.232 | 0.671 | -7.795 | <0.0001 |
| Time (year) | 1.425 | 0.349 | 4.071 | 0.0036 |
| Time > 2014 (year) | -1.435 | 0.358 | -4.003 | 0.0039 |

[18]. The frequency of cases was highest in the districts of Kalutara (118) and Colombo (108) in Western Province. However, the case distribution per population was highest for Matara, and Puttalam districts.

Previous studies have examined the impact of MDA on filarial lymphoedema with variable results, some reporting reductions in chronic manifestations while some observed reductions in acute manifestations [19–25]. Studies conducted prior to the elimination drive utilized different treatment regimens from those utilized in the global program for the elimination of LF. Reductions in MF rates, elephantiasis and hydroceles were reported in Tahiti following five years of standard DEC therapy combined with vector control, in Indonesia 11 years of MDA with DEC reported reductions in elephantiasis, acute attacks and MF rates while in China six months of DEC medicated table salt reduced acute manifestations but chronic manifestations were unchanged or aggravated [19–21]. Similarly, studies conducted after the global elimination drive also report mixed results [22–25]

In the present evaluation, the new lymphoedema case visits presenting for MMDP services nearly sixteen years after five rounds of MDAs show a 70% reduction from 2 per 10,000 populations in 2006 following the fifth round of preventive chemotherapy to 0.6 per 10,000 population in 2022. The average infection parameters, MF and infective mosquito rates have shown a steady decline to its current low value of 0.03% for both indices suggesting that Sri Lanka is nearing transmission breakpoints with eventual cessation of transmission (risk-of-infection

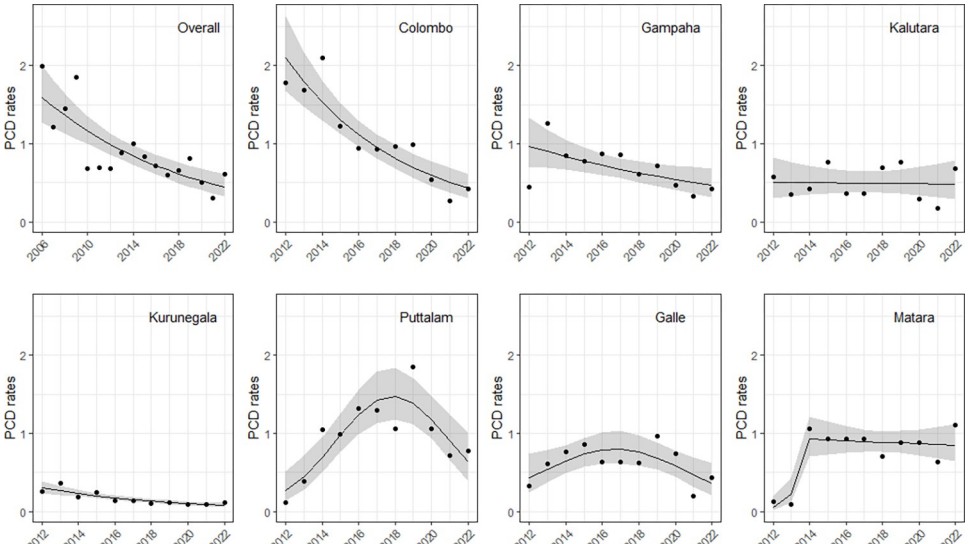

**Fig 2. Observed and predicted overall passive case detection rates from 2006 to 2022 and district-wise rates from 2012 to 2022.** Observed PCD rates (black dots) and predicted PCD rates (black lie) with 95% Confidence Interval (grey colour band) from the fitted log-linear models.

nearing zero) [8]. The overall declining trends in new lymphoedema case detection rates provide supportive evidence. However, district-wise data indicate that the declining trend is not uniform throughout the endemic region. Declining trends in PCD rates were observed in three districts, Colombo, Gampaha and Kurunegala. The district of Kurunegala was a low transmission region in the pre-elimination era and five rounds of MDAs would have sufficed in interrupting transmission. The district of Colombo (where the capital city is located) and the adjacent district Gampaha, probably experienced more socioeconomic and infrastructure development than the other regions thus lowering vector densities and transmission. In the other four districts a steady decline was not evident. Thus, a careful epidemiological assessment of these four districts may be a worthwhile effort.

The disease dynamics of LF is an important aspect that needs to be considered as the appearance of chronic manifestations such as lymphoedema occurs about a decade after the infection. Thus, the disease rates reflect the infection dynamics a decade ago. Therefore, historical surveillance data prior to validation was important. The southern districts of Matara and Galle were high LF endemic districts in the pre-elimination era with an average MF prevalence of 4.4% and MF density of 20 /60μl of blood [26]. Even after cessation of the MDA (nine years post-MDA), there was strong evidence of persistent transmission, with vector infection rates by molecular xenomonitoring (filarial DNA in *C. quinquefasciatus*, principal vector of *W. bancrofti* in Sri Lanka) above the threshold value of 1% [27].

The emergence of brugian filariasis infections of zoonotic origin with the district of Puttalam being the epicenter of the outbreak was reported about a decade ago [28,29]. The entomological surveillance data in the district of Puttalam support ongoing transmission [10]. As of now, the emerging brugian filariasis infection has spread to other endemic districts particularly Kalutara, and Galle. Therefore, it may be presumed that evaluation may be premature in the districts of Puttalam and Kalutara (reemerged brugia infections), Matara and Galle (former high transmission regions) to observe a steady declining trend in lymphatic morbidity rates.

Thus the variable decline in PCD rates of lymphoedema observed across the endemic region may be due to the heterogeneity in transmission. Factors such as low baseline MF

**Table 3. Age distribution of new lymphoedema patient visits to filaria clinics by districts (2019–2022).**

| Age category by district | 2019 | 2020 | 2021 | 2022 |
|---|---|---|---|---|
| **Colombo** | | | | |
| Mean Age (years) | 56.2 | 52.7 | 52.2 | 55.4 |
| 0–20 years | 9 | 3 | 6 | 0 |
| 21–40 years | 27 | 24 | 7 | 11 |
| 41–60 years | 99 | 61 | 29 | 54 |
| 61–80 years | 113 | 38 | 20 | 37 |
| 81–100 years | 5 | 5 | 3 | 1 |
| **Gampaha** | | | | |
| Mean Age (years) | 57.6 | 51.1 | 49.8 | 53.1 |
| 0–20 years | 2 | 3 | 2 | 1 |
| 21–40 years | 19 | 31 | 23 | 19 |
| 41–60 years | 64 | 37 | 30 | 31 |
| 61–80 years | 86 | 39 | 26 | 30 |
| 81–100 years | 1 | 2 | 0 | 2 |
| **Kalutara** | | | | |
| Mean Age (years) | 56.4 | 52.6 | 54.5 | 58.4 |
| 0–20 years | 4 | 1 | 0 | 0 |
| 21–40 years | 9 | 3 | 3 | 11 |
| 41–60 years | 44 | 25 | 11 | 31 |
| 61–80 years | 49 | 8 | 8 | 44 |
| 81–100 years | 1 | 1 | 0 | 2 |
| **Galle** | | | | |
| Mean Age (years) | 52.2 | 50.7 | 53.2 | 51.6 |
| 0–20 years | 9 | 4 | 2 | 4 |
| 21–40 years | 16 | 18 | 3 | 5 |
| 41–60 years | 40 | 35 | 9 | 24 |
| 61–80 years | 42 | 25 | 11 | 17 |
| 81–100 years | 2 | 2 | 0 | 0 |
| **Matara** | | | | |
| Mean Age (years) | 57.9 | 60.7 | 50.7 | 50.8 |
| <20 years | 2 | 10 | 3 | 7 |
| 21–40 years | 6 | 24 | 9 | 18 |
| 41–60 years | 29 | 21 | 27 | 37 |
| 61–80 years | 38 | 21 | 15 | 34 |
| 81–100 years | 1 | 0 | 1 | 1 |
| **Kurunegala** | | | | |
| Mean Age (years) | 60.0 | 35.3 | 32.7 | 60.8 |
| <20 years | 0 | 4 | 4 | 0 |
| 21–40 years | 2 | 3 | 5 | 2 |
| 41–60 years | 7 | 8 | 6 | 10 |
| 61–80 years | 10 | 0 | 0 | 12 |
| 81–100 years | 1 | 0 | 0 | 2 |
| **Puttalam** | | | | |
| Mean Age (years) | 54.2 | 50.7 | 45.1 | 49.2 |
| <20 years | 10 | 10 | 4 | 3 |
| 21–40 years | 16 | 14 | 20 | 16 |
| 41–60 years | 62 | 29 | 24 | 39 |

(*Continued*)

**Table 3.** (Continued)

| Age category by district | 2019 | 2020 | 2021 | 2022 |
|---|---|---|---|---|
| 61–80 years | 64 | 35 | 13 | 17 |
| 81–100 years | 2 | 1 | 0 | 1 |
| **Sri Lanka** | | | | |
| Mean Age (years) | 55.9 | 50.0 | 49.2 | 53.6 |
| <20 years | 36 | 35 | 21 | 15 |
| 21–40 years | 95 | 117 | 70 | 87 |
| 41–60 years | 345 | 216 | 136 | 244 |
| 61–80 years | 402 | 166 | 91 | 198 |
| 81–100 years | 13 | 11 | 4 | 9 |

prevalence prior to MDA (district of Kurunegala) and socioeconomic growth leading to declines in vector densities (district of Colombo and Gampaha) would have contributed to the steady decline in morbidity while persistence of bancroftian filariasis in small pockets of high baseline prevalence regions (districts of Matara and Galle), emergence of zoonotic brugian filariasis (Kalutara and Puttalam) are challenges to declines in morbidity rates while non-infection parameters such as an improved referral rates to filarial clinics may also have contributed to maintenance or rising trends in morbidity.

The explanations for new lymphoedema case detections after elimination of LF as a public health problem may vary according to region. Some cases may represent late manifestations of asymptomatic infections, some may represent early manifestations of newly acquired infections of bancroftian (southern districts of Matara and Galle) or brugian filariasis (districts of Puttalam and Kalutara) while some lymphoedema cases may be of non-filarial origin (primary lymphoedema or other secondary cause). The latter two may explain the occurrence of new cases among the younger population (<20 years).

Unlike the lymphatic pathology, the presence of adult worms alone is sufficient to cause hydroceles [30]. Thus, the reduction of adult worm burden with repeated MDAs with anti-filarial drugs which have a partial adulticidal effect should reduce the hydrocele prevalence. However, the hospital admission rates for hydrocele management remained static until 2015 (nearly a decade following conclusion of MDA) and only thereafter a declining trend was evident. (Fig 3). The delayed decline observed may have been caused by the backlog clearance of hydrocele/ spermatocele surgeries. A review of the literature reports that four rounds of MDA with either DEC/ivermectin achieved a maximum of 60% reduction in hydrocele prevalence and beyond that there was no additional impact [31].

This report provides a glimpse of the lymphatic filariasis morbidity trends, two decades following the NPELF consisting of five rounds of MDAs with DEC and albendazole in an area endemic to bancroftian filariasis and a re-emerging zoonotic *B. malayi*. It is timely to ascertain the national case burden of lymphoedema and hydrocele by active surveillance and to monitor

**Table 4. Parameter estimates of the fitted log-linear model for hospital admission rate for the management of hydroceles/spermatoceles.**

| | Estimate | Std. Err. | T value | P value |
|---|---|---|---|---|
| Overall | | | | |
| Intercept | 16.517886 | 17.949465 | 0.920 | 0.37421 |
| Time (year) | -0.008876 | 0.008924 | -0.995 | 0.33805 |
| Time > 2015 (year) | -0.075336 | 0.018445 | -4.084 | 0.0012 |

Observed hospital admission rates (black dots) and predicted rates (black lie) with 95% Confidence Interval (grey colour band) from the fitted log-linear models.

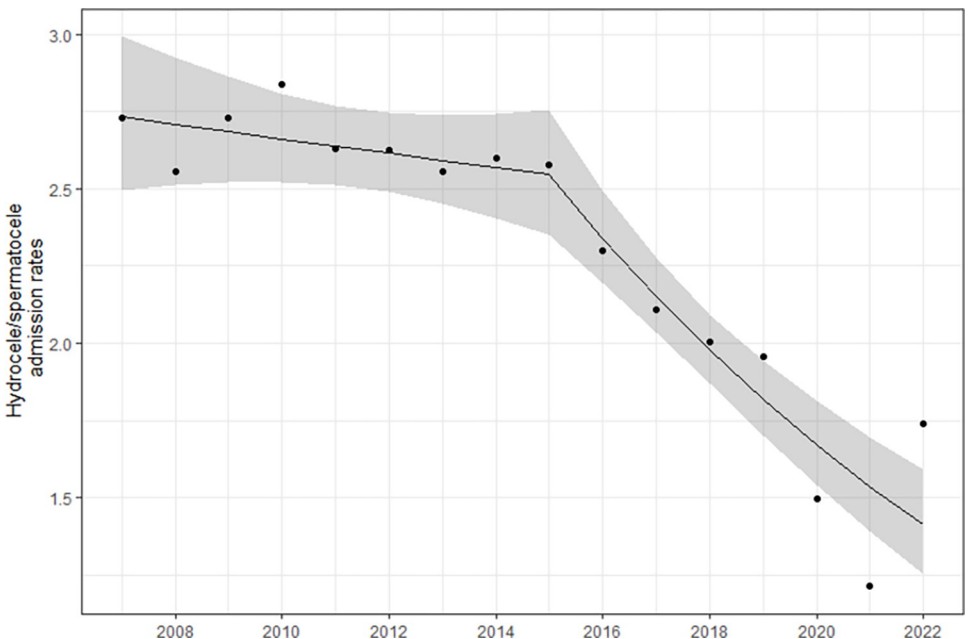

**Fig 3. Hospital admission rates for management of hydrocele/spermatocele in Sri Lanka following the mass drug administration program (2007–2022).**

the trends in disease rates, particularly in the districts showing static trends or marginal non-significant declines in PCD rates. The factors discussed herein may be contributing factors for the variable declining trends in chronic filariasis morbidity observed in the country. Other factors such as hygiene-related factors (environmental and personal) and genetic predisposition need consideration and rectification where possible.

Factors such as lack of confidence or excessive enthusiasm in the newly established MMDP services may have affected clinic attendance at the onset or later which may affect the PCD rates and trend lines. The absence of baseline data was a major limitation in the assessment.

## Conclusions

The chronic lymphatic disease rates in the LF endemic region in Sri Lanka, ascertained by PCD, showed a steady decline in the overall endemic region, and, specifically in Colombo, Gampaha and Kurunegala districts. Except for the district of Kalutara, where the lymphoedema PCD rate remained static, all other districts showed declining trends in PCD rates in recent years. Hydrocele/ spermatocele case admission rates to state hospitals showed a downward trend from 2015 onwards, about a decade following the conclusion of MDA. Ascertaining the actual lymphoedema case burden in the country (endemic and non-endemic regions) and monitoring disease trends is required at this juncture of the near elimination of transmission of LF in Sri Lanka. Integration of morbidity management services to the general health system and enhancing services with the inclusion of tertiary care (rehabilitation, psychological and social support), where indicated, is suggested with the reductions in the public health burden of lymphatic filariasis.

## Supporting information

**S1 Data. New lymphoedema case numbers registered in filarial clinics district wise from 2012–2022 with the estimated population for the relevant years.**
(CSV)

**S2 Data. Patient numbers admitted to state hospitals in Sri Lanka for management of hydrocele/spermatocoeles from 2007–2022 with the estimated mid-year male population.** (CSV)

## Acknowledgments

The authors gratefully acknowledge the dedication of the Filaria clinic staff towards data maintenance while providing morbidity prevention and alleviation services to patients with lymphoedema and the Directors of the Anti-Filariasis Campaign, for their leadership, guidance and dedication for lymphatic filariasis elimination. We also acknowledge the support extended by the Medical Statistics Division of the Ministry of Health.

## Author Contributions

**Conceptualization:** Indeewarie E Gunaratna, Nilmini T. G. A Chandrasena, Ranjan Premaratna.

**Data curation:** Indeewarie E Gunaratna, Murali Vallipuranathan.

**Formal analysis:** Indeewarie E Gunaratna, Nilmini T. G. A Chandrasena, Dileepa Ediriweera.

**Investigation:** Indeewarie E Gunaratna, Nilmini T. G. A Chandrasena, Murali Vallipuranathan.

**Methodology:** Nilmini T. G. A Chandrasena.

**Supervision:** Nilanthi R de Silva.

**Writing – original draft:** Nilmini T. G. A Chandrasena, Ranjan Premaratna.

**Writing – review & editing:** Dileepa Ediriweera, Nilanthi R de Silva.

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
