## [Decision Letter · Decision Letter 0]

31 Mar 2024

Dear prof Chandrasena,

Thank you very much for submitting your manuscript "The impact of the National Programme to Eliminate Lymphatic Filariasis on filariasis morbidity in Sri Lanka:  comparison of current status with retrospective data following the elimination of lymphatic filariasis as a public health problem." for consideration at PLOS Neglected Tropical Diseases. As with all papers reviewed by the journal, your manuscript was reviewed by members of the editorial board and by several independent reviewers. In light of the reviews (below this email), we would like to invite the resubmission of a significantly-revised version that takes into account the reviewers' comments. 

We cannot make any decision about publication until we have seen the revised manuscript and your response to the reviewers' comments. Your revised manuscript is also likely to be sent to reviewers for further evaluation.

Sincerely,

Wilma A. Stolk, Ph.D.

Academic Editor

Richard Bradbury

Section Editor

Reviewer's Responses to Questions

**Key Review Criteria Required for Acceptance?**

**Methods**

-Are the objectives of the study clearly articulated with a clear testable hypothesis stated?

-Is the study design appropriate to address the stated objectives?

-Is the population clearly described and appropriate for the hypothesis being tested?

-Is the sample size sufficient to ensure adequate power to address the hypothesis being tested?

-Were correct statistical analysis used to support conclusions?

-Are there concerns about ethical or regulatory requirements being met?

Reviewer #1: Yes to all, but suggestion for a detailed analysis is given in the attached review report.

Reviewer #2: Objective is relatively clear however the methodology needs improving. 

The study design is not clear. It does not provide rational for the different years of hydrocoele and lymphoedema cases collected. 

Population estimates are based on the 2012 census. It does not seem to have taken population growth into account over the long time period. Adjustments need to be considered. 

More clarity on the hydrocoele admission process, it is free for all patients, do the have access, waiting lists. Why were hospital data not included to 2022?

No mention of how reporting systems may have changed over time - this could influence the results.

No mention of ethics approval for accessing clinic or hospital records.

Reviewer #3: - Objectives stated clearly

- If you have data on population and prevalence, please consider to add DALYs.

- Unclear how many hospitals/where and if representative sample for the whole of Sri Lanka. So unclear if population is appropriate. 

- Are there no updated census data available? This has implications for the results, need to discuss in the Discussion

- How to ensure anonymity of patient records collected for the study?

- Why hydrocele data only till 2017? This is a great limitations of this study. Are the patient database comparable between lymphoedema and hydrocele?

**Results**

-Does the analysis presented match the analysis plan?

-Are the results clearly and completely presented?

-Are the figures (Tables, Images) of sufficient quality for clarity?

Reviewer #1: (1) Yes but needs clarity on the methods used for statistical Analysis.

(2) Results are clear and presented well, more results based on detailed analysis is suggested.

(3) Figures are with good quality and useful for interpretation.

Also, please see the attached review report for a detailed analysis and presentation of results.

Reviewer #2: The case numbers are presented for 2022 – are there similar numbers and demographic information for the other years? The difference in average age in 2006 to 2022 would be informative. 

The figures' resolution could be improved . All data in the figures and tables need to be included as supplementary files.

Reviewer #3: - How come that there are so many new cases of morbidity among young people (<21 years)? Does this mean that there is still circulating infection?

- Please add a geographic map with per region the levels/categories of clinical morbidity over time periods, such as to see straight in which districts potential transmission still occurs, and where more MMDP needs to be invested.

- Provide (%) in Table 1, to compare variables

- Actually only Colombo (most cases), Kurunegala, and Gampaha are declining in cases. This is an interesting result, and it needs to be described more clearly why no decline is seen in the other areas. What challenges contribute to a lack in decline?

- Sri Lanka overall results are mostly driven by Colombo, and this needs to be stated somewhere. There actually is no country-wide decline. 

- Are there no regional figures to be made for hydrocele?

**Conclusions**

-Are the conclusions supported by the data presented?

-Are the limitations of analysis clearly described?

-Do the authors discuss how these data can be helpful to advance our understanding of the topic under study?

-Is public health relevance addressed?

Reviewer #1: Yes to all

Reviewer #2: The discussion needs to highlight more on why there are differences in the rates of decline or why there is no decline in the number of cases - i..e what are potential reasons - socio-economic improvement in Colombo may have important, or in another areas the evidence of ongoing transmission may have helped to maintain case numbers.

Overall referencing needs to be improved.

Seems more focus on transmission than morbidity.

Reviewer #3: - Very long discussion. Needs to be more comprehensive, and to the point. 

- If Tx was provided, you expect a halt in progression of tissue damage. Thus a halt in development of clinical disease. Still, in this study, the authors report new clinical disease, primarily among the young population. I think this data must be coupled with (historical) MF prevalence data to get a better understanding of why we see these new lymphoedema cases arising. 

- Very long description of what is found in other countries (lines 215-232), but we need to get a grip of what happens in Sri Lanka, which is missing here. 

- Lines 236-239, infection parameters are mentioned, but not shown, whereas this data is critical to understand the results of this study

**Editorial and Data Presentation Modifications?**

Reviewer #1: (No Response)

Reviewer #2: (No Response)

Reviewer #3: Overall prospects of an interesting paper, and great to see the progress Sri Lanka is making. 

- Create some space between x-axis and y-axis and legends in the figures. 

- Overall pixels are very low of the figures. Please submit again with improved pixel-level. 

- R-squared value is often overwriting the histograms of the figures

**Summary and General Comments**

Reviewer #1: This manuscript aims to assess the impact of five rounds of MDA with DA (2002-2006) on LF disease manifestations (lymphoedema and hydrocele) sixteen years after stopping MDA. The authors using data on passive case detection on lymphoedema (from filariasis clinics) and hospital admission on hydrocele (hospital records) explored the trends in (i) passive case detection (PCD) rates (ratio of the no. of cases detected to that of estimated population in 2022) for lymphoedema by district (2012-2022) and overall (from 2006 -2022), and (ii) hydrocele admission rates from 2007-2017 by district / province and overall. Globally, very few studies have examined the impact of MDA on LF chronic disease manifestations and the results are highly variable. As WHO is targeted to achieve LF elimination by 2030, the results of this study are important to know the status of global effort to eliminate LF as a public health problem. 

The manuscript is well written with clear background and objectives. The results are presented well with figures and Tables, and the discussion is narrated comparing results available from studies elsewhere. While I recommend the ms for publication, I would request the authors to revise ms with results based on a detailed analysis of this historical data on clinical manifestations, before accepting for publication.

Reviewer #2: The paper's aim of looking at changes in case numbers over time is an important one, however, the methods and results need to be improved. The paper seems a little superficial. Language and grammar needs improving. Some references quite old/outdated. e.g. ref #10 is old to refer to inconsistency in numbers or just lacking in relation to big statements e.g. line 290. Inconsistencies with abbreviation usage - LF vs. lymphatic filariasis

Introduction 

- The clinical manifestations need to be described better

- It would be useful to have a map of the endemic districts and provinces

- The overall rate of MF in 2001 is very low – what was the variability/range across the endemic districts at this time?

- What does the post-validation surveillance entail – a brief description would be helpful.

- The statement that LF morbidity has not been assessed doesn’t seem to take other pieces of work into account - There are a few papers that refer to the burden and also morbidity e.g. https://pubmed.ncbi.nlm.nih.gov/18060080/
https://pubmed.ncbi.nlm.nih.gov/16004709/
https://pubmed.ncbi.nlm.nih.gov/29175490/ The lack of references to other morbidity papers in Sri Lanka suggest the co-authors need to do more background reading.

How has reporting systems changed / if they have changed

Reviewer #3: - Interesting paper and nice that they looked at clinical morbidity. Overall prospects of an interesting paper, and great to see the progress Sri Lanka is making. 

- Why mention R2 in abstract if not explained. Also no hydrocele rates mentioned in the abstract. Need to use same approach of reporting for lymphoedema and hydrocele. 

- Hydrocele data and analyses are unfortunately very limited and difficult to compare with the lymphoedema data (other years, other hospitals?, no regional data, etc)

- Please add the database as a Supplement

PLOS authors have the option to publish the peer review history of their article (what does this mean?). If published, this will include your full peer review and any attached files.

Reviewer #1: No

Reviewer #2: No

Reviewer #3: No
---

## [Decision Letter · Decision Letter 1]

6 Jul 2024

Dear prof Chandrasena,

We are pleased to inform you that your manuscript 'The impact of the National Programme to Eliminate Lymphatic Filariasis on filariasis morbidity in Sri Lanka:  comparison of current status with retrospective data following the elimination of lymphatic filariasis as a public health problem.' has been provisionally accepted for publication in PLOS Neglected Tropical Diseases.

Best regards,

Jong-Yil Chai

Section Editor

---

## [Editor Report · Acceptance letter]

25 Jul 2024

Dear prof Chandrasena,

We are delighted to inform you that your manuscript, "The impact of the National Programme to Eliminate Lymphatic Filariasis on filariasis morbidity in Sri Lanka:  comparison of current status with retrospective data following the elimination of lymphatic filariasis as a public health problem.," has been formally accepted for publication in PLOS Neglected Tropical Diseases.

Best regards,

Shaden Kamhawi

co-Editor-in-Chief

Paul Brindley

co-Editor-in-Chief
